# Analysis of *FBN1*, *TGFβ2*, *TGFβR1* and *TGFβR2* mRNA as Key Molecular Mechanisms in the Damage of Aortic Aneurysm and Dissection in Marfan Syndrome

**DOI:** 10.3390/ijms26073067

**Published:** 2025-03-27

**Authors:** María Elena Soto, Myrlene Rodríguez-Brito, Israel Pérez-Torres, Valentín Herrera-Alarcon, Humberto Martínez-Hernández, Iván Hernández, Vicente Castrejón-Téllez, Betsy Anaid Peña-Ocaña, Edith Alvarez-Leon, Linaloe Manzano-Pech, Ricardo Gamboa, Giovanny Fuentevilla-Alvarez, Claudia Huesca-Gómez

**Affiliations:** 1Research Direction, Instituto Nacional de Cardiología Ignacio Chávez, Juan Badiano No. 1, Col. Sección XVI, Mexico City 14080, Mexico; elena.soto@cardiologia.org.mx; 2Cardiovascular Line in American British Cowdray (ABC) Medical Center, PAI ABC Sur 136 No. 16, Col. Las Américas, Mexico City 01120, Mexico; 3Cardiothoracic Surgery Department, Instituto Nacional de Cardiología Ignacio Chávez, Juan Badiano No. 1, Col. 4 Sección XVI, Mexico City 14080, Mexico; dra.rodriguez.brito.myrlene@gmail.com (M.R.-B.); cardiosure@gmail.com (V.H.-A.); humbertomartinez@hotmail.com (H.M.-H.); ihm56@hotmail.com (I.H.); 4Cardiovascular Biomedicine Department, Instituto Nacional de Cardiología Ignacio Chávez, Juan Badiano No. 1, Col. Sección XVI, Mexico City 14080, Mexico; pertorisr@yahoo.com.mx (I.P.-T.); loe_mana@hotmail.com (L.M.-P.); 5Physiology Department, Instituto Nacional de Cardiología Ignacio Chávez, Juan Badiano No. 1, Col. 4 Sección XVI, Mexico City 14080, Mexico; vcastrejn@yahoo.com.mx (V.C.-T.);; 6Biochemistry Department, Instituto Nacional de Cardiología Ignacio Chávez, Juan Badiano No. 1, Col. Sección XVI, Mexico City 14080, Mexico; betsy.anaid@gmail.com; 7Tecnológico Nacional de México, Instituto Tecnológico de Tuxtla Gutiérrez, Tuxtla Gutiérrez 29050, Mexico; 8Sub-Directorate of Basic Research, Instituto Nacional de Cardiología Ignacio Chávez, Juan Badiano No. 1, Col. Sección XVI, Mexico City 14080, Mexico; edith.alvarez@cardiologia.org.mx; 9Endocrinology Department, Instituto Nacional de Cardiología Ignacio Chávez, Juan Badiano No. 1, Col. 4 Sección XVI, Mexico City 14080, Mexico

**Keywords:** Marfan syndrome, aortic dissection, *FBN1*, *TGFβR1*, *TGFβR2*, *TGFβ2*

## Abstract

Marfan syndrome (MFS) is an inherited connective tissue disorder, with aortic root aneurysm and/or dissection being the most severe and life-threatening complication. These conditions have been linked to pathogenic variants in the *FBN1* gene and dysregulated *TGFβ* signaling. Our objective was to evaluate the mRNA expression of *FBN1*, *TGFBR1*, *TGFBR2*, and *TGFB2* in aortic tissue from MFS patients undergoing surgery for aortic dilation. This prospective study (2014–2023) included 20 MFS patients diagnosed according to the 2010 Ghent criteria, who underwent surgery for aneurysm or dissection based on Heart Team recommendations, along with 20 non-MFS controls. RNA was extracted, and mRNA levels were quantified using RT-qPCR. Patients with dissection showed significantly higher *FBN1* mRNA levels [79 (48.1–110.1)] compared to controls [37.2 (25.1–79)] (*p* = 0.03). Conversely, *TGFB2* expression was significantly lower in MFS patients [12.17 (6.54–24.70)] than in controls [44.29 (25.85–85.36)] (*p* = 0.029). A positive correlation was observed between higher *FBN1* expression and a larger sinotubular junction diameter (r = 0.42, *p* = 0.07), while increased *FBN1* expression was particularly evident in MFS patients with dissection. Additionally, *TGFB2* expression showed an inverse correlation with ascending aortic diameter (r = 0.53, *p* = 0.01). In aortic tissue, we found decreased *TGFB2* and receptor levels alongside increased *FBN1* mRNA levels. These molecular alterations may reflect compensatory mechanisms in response to tissue damage caused by mechanical stress, leading to dysregulation of physiological signaling pathways and ultimately contributing to aortic dilation in MFS.

## 1. Introduction

Marfan syndrome (MFS) is a hereditary connective tissue disorder [1]. Its incidence varies, but the estimated prevalence is approximately 1 in 10,000. The condition has high penetrance with variable clinical expression.

The clinical criteria for diagnosing MFS are based on its effects on the most commonly affected tissues, including the musculoskeletal system, eyes, cardiovascular and pulmonary tissues, and skin [2]. One of the most serious and potentially life-threatening complications is aortic root aneurysm, which can lead to dissection and rupture. Other key clinical features include mitral valve prolapse, pneumothorax, dural ectasia, and myopia. A defining characteristic of the affected tissues is their high fibrillin content, particularly in the aorta, musculoskeletal structures, lungs, and cornea.

Fibrillin-1 (FBN1) plays a critical role in TGFβ2 signaling, and genetic variants in *FBN1* lead to increased bioavailability and activity of TGFβ2. Pathogenic variants in the *FBN1* gene, which encodes the FBN1 protein, have been identified as the primary cause of MFS. FBN1 is the main component of 10–12 nm microfibrils in the extracellular matrix of connective tissue. Most pathogenic variants in *FBN1* are missense mutations or nonsense mutations that introduce premature termination codons [3]. These variants can lead to the production of truncated *FBN1* proteins that may still be incorporated into extracellular microfibrils. However, truncated transcripts are typically degraded by nonsense-mediated mRNA decay. The resulting loss of elastic fiber integrity weakens the vessel walls, reducing their ability to withstand mechanical stress [4].

Additionally, a significant reduction in aortic contraction has been observed, suggesting potential mechanisms involved in the development and progression of aortic aneurysms in Marfan syndrome (MFS). This may contribute to the deterioration of the wall’s mechanical properties and structural integrity. While aneurysm formation appears to have a genetic basis as a triggering factor, other elements, such as mechanical stress and inflammatory processes, may also play a role [5]. Moreover, MFS patients exhibit markedly lower *TGFβ2* mRNA levels, potentially indicating an inflammatory response that further weakens the aortic wall and accelerates its degeneration [6].

Different pathogenic variants in *TGFβR1* and *TGFβR2*, receptors for transforming growth factor beta (TGFβ), have been identified [7]. Pathogenic variants in these genes were later recognized as the cause of a distinct condition, now classified as Loeys–Dietz syndrome [8]. While the idea that abnormally activated TGFβ signaling is responsible for Marfan syndrome and related disorders remains uncertain, it is evident that FBN1 and components of the TGFβ signaling pathway, including *TGFβR1* and *TGFβR2*, share overlapping genetic mechanisms [4,7].

TGFβ2 first binds to its type 2 receptor (TGFβR2) on the cell surface, which subsequently recruits and phosphorylates the type 1 receptor (TGFβR1). Upon activation, TGFβR1 phosphorylates the SMAD proteins (SMAD2 and SMAD3).

The phosphorylated SMAD2 and SMAD3 then bind to SMAD4 and are transported into the nucleus with the help of importins 7 and 8 [9]. Inside the nucleus, they regulate the expression of TGFβ target genes by interacting with transcription factors and cofactors, including coactivators and corepressors [10,11,12]. TGFβ2 can also signal through non-SMAD (non-canonical) pathways, such as those involving mitogen-activated protein kinases (MAPK) [13,14]. Recent studies support the hypothesis that TGFβ signaling promotes the development of thoracic aortic aneurysms (TAAs). This has been demonstrated in animal models and is further supported by findings in human aortas with advanced TAA [15]. While the increase in circulating TGFβ within the extracellular matrix has been widely studied, little is known about whether its expression is also elevated intracellularly in response to genetic or epigenetic modulation in both canonical and non-canonical signaling pathways. Therefore, our objective was to assess the mRNA expression of *FBN1*, *TGFBR1*, *TGFBR2*, and *TGFB2* in the aortic tissue of Marfan syndrome patients with aortic dilation.

## 2. Results

### 2.1. Characteristics of the Study Population

Table 1 compares demographic characteristics and laboratory findings between patients with Marfan syndrome (MFS) and controls. The study included 40 participants, evenly divided between the MFS group and controls. The MFS group had a significantly younger average age (32 ± 8 years) compared to controls (64 ± 13 years) (*p* = 0.0001). Regarding medical conditions, systemic arterial hypertension (SAH) was less prevalent in the MFS group (25%) compared to controls (70%) (*p* = 0.0001).

The characteristics of the total Ghent criteria are shown in Table 2. Ten patients (50%) had no known family history of Marfan syndrome (MFS) or aortic dissection, while the other ten (50%) had a family history of both. All patients met at least two criteria; 7 (35%) met two criteria, 7 (35%) met three, 5 (25%) met four, and 1 (5%) met five.

The genetic variants identified in FBN1 in five patients follow the Human Genome Variation Society (HGVS) nomenclature. Patient 2 harbors NM_000138.5:c.3697C>T (p.Gln1233Ter), a stop-gain mutation. Patient 4 has NM_000138.5:c.2258G>T (p.Gly753Val), a non-synonymous SNV. Patient 11 has NM_000138.5:c.1539T>G (p.Cys513Trp), a non-synonymous SNV. Patient 12 carries NM_000138.5:c.4621C>T (p.Arg1541Ter), a stop-gain variant. Patient 14 presents NM_000138.5:c.7133G>A (p.Cys2378Tyr), a non-synonymous SNV.

A comparison of surgical conditions and procedures between MFS patients and controls is presented in Table 3. The study included 40 patients, 20 with MFS and 20 controls. Aortic dilation was significantly more frequent in MFS patients (85%) compared to controls (45%), with *p* = 0.01. Aortic dissection occurred in 17.5% of the total population, affecting 10% of MFS patients and 25% of controls. Clamping time was notably longer in MFS patients (median 152 min, range 110–224) compared to controls (median 131 min, range 66–254), with *p* = 0.03. Lastly, mortality was significantly higher in the control group (20%) compared to the MFS group (0%), with *p* = 0.05.

Seven patients with MFS stopped attending the institution. We contacted them by phone, confirmed they were alive, and invited them to return for follow-up. In the control group, two patients stopped attending. One was successfully located and confirmed to be alive. The other was also alive, but we could not contact him directly. Many patients were from other countries and did not provide contact information, making follow-up difficult.

### 2.2. Surgical Characteristics 

Table 4 summarizes the surgical characteristics and procedures performed. Measurements of the aortic ring diameter (ARD) and sinus of Valsalva diameter (SVD) demonstrated considerable variability, with ARD ranging from 22 mm to 62 mm and SVD from 26 mm to 94 mm. Similarly, the diameter of the sinotubular junction (STJ) ranged from 26 mm to 100 mm.

The most frequently performed surgical procedures included the Bentall technique, undertaken in 12 patients, and aortic replacement, conducted in 4 patients. Less common interventions included supra-aortic trunk revascularization and aortic valve resuspension (David technique), each performed in a single patient. Additionally, combined procedures were performed in 3 patients, such as the Bentall technique combined with tricuspid valve repair and surgical packing.

A comparative analysis of aortic diameters between patients with Marfan syndrome (MFS) and controls revealed significant differences in key measurements. The aortic root diameter was significantly larger in the MFS group (37.55 ± 7.5 mm) compared to controls (30.8 ± 6.1 mm; *p* = 0.004). Similarly, the sinus of Valsalva and sinotubular junction diameters were markedly increased in MFS patients (61.3 ± 20.9 mm and 57.1 ± 21.4 mm, respectively) compared to controls (38.4 ± 5.2 mm and 38.2 ± 9.9 mm, respectively; *p* = 0.0001 for both). While the ascending aorta diameter was larger in MFS patients (58.8 ± 24.3 mm) than in controls (47.4 ± 15.5 mm), this difference did not reach statistical significance (*p* = 0.08). Detailed comparisons are presented in Table 5.

Figure 1 illustrates the histological findings in MFS patients with aortic pathology. Panel A shows a representative photomicrograph of a 28-year-old male with a giant ascending aortic aneurysm, revealing ruptured elastic fibers in the adventitia and minimal inflammatory infiltrate, stained with hematoxylin and eosin (H&E). Panel B depicts a 36-year-old female with Stanford type A aortic dissection extending into the supra-aortic trunks, highlighting the loss of elastic fiber continuity, cystic medial necrosis, vascular medial scarring, wall dissection, and focal thrombosis, stained with Weigert’s method for elastic fibers. Panel C presents findings from a 33-year-old male with DeBakey type II aortic dissection, showing extensive destruction of elastic fibers and cystic medial necrosis, stained with Masson’s trichrome.

### 2.3. FBN1, TGFβR1, TGβR2 and TGFβ2 mRNA Level Expression

With the purpose of knowing the expression levels of the genes studied and knowing if there are differences between the values with respect to the patients considered as controls, the measurement was carried out in aortic tissues of our study population. Figure 2 shows the expression levels of mRNA for *FBN1*, *TGFβR1*, *TGFβR2*, and *TGFβ2*, evaluated in both groups, comparing individuals with Marfan syndrome to the control group. The mRNA levels were compared between controls and Marfan patients, both with and without aortic dilation, aneurysm, or dissection; see Table 6.

For *FBN1*, MFS patients with dissection (n = 4) had levels of 79 (48.1–110.1), compared to controls with dissection (n = 3), who had levels of 37.2 (25.1–79), with (*p* = 0.03). For *TGFβR1* mRNA levels, we found no differences between MFS and controls who had dilation or dissection.

For *TGFβR2* mRNA levels in patients with dilatation, we observed [Control 96.3 (0.05–282.02) vs. MFS [4 (0.40–41.3) *p* = 0.05]. In the case of the *TGFβ2* mRNA in dissection, MFS patients had markedly lower levels of *TGFβ2*, with a median of 12.17 (6.54–24.70) vs. controls 44.28 (25.85–85.36), with (*p* = 0.02).

The results presented in Table 7 show the correlation between mRNA expression levels of *FBN1*, *TGFβR1*, *TGFβR2*, and *TGFβ2* and different aortic segments for both Marfan patients and controls. Pearson correlation coefficients (r) and *p*-values are provided for each aortic segment, including the aortic root, sinus of Valsalva, sinotubular junction, and ascending aorta. In patients with Marfan syndrome, a positive correlation was observed between mRNA *FBN1* expression and the diameter of the sinotubular junction (r = 0.42, *p* = 0.07), as well as between mRNA *TGFβ2* expression and the diameter of the ascending aorta (r = 0.53, *p* = 0.01), suggesting a possible relationship between these molecular markers and aortic enlargement in these patients. In the control group, a positive and significant correlation was detected between mRNA *TGFβ2* expression and the diameter of the aortic root (r = 0.62, *p* = 0.004), as well as between mRNA *TGFβR1* expression and the diameter of the ascending aorta (r = 0.81, *p* = 0.001). Additionally, mRNA *TGFβ2* expression showed a significant negative correlation with the diameter of the sinus of Valsalva (r = −0.46, *p* = 0.039).

To evaluate whether the presence of FBN1 variants influences the expression of genes related to the TGFβ2 pathway, patients with Marfan syndrome were stratified into two groups: those with identified FBN1 mutations (MFS-Variants) and those without genetic testing (MFS-TNP). Both groups were compared to healthy controls.

Table 8 show a significant reduction in *TGFβR2* expression in MFS-Variants compared to controls [Control 67.86 (0.05–282.02) vs. MFS-Variants 2.03 (0.03–3.61), *p* = 0.02]. Similarly, MFS-TNP also exhibited lower *TGFβR2* expression than controls [4.03 (0.34–25.62), *p* = 0.01]; however, no significant difference was observed between MFS-Variants and MFS-TNP (*p* = 0.16).

## 3. Discussion

For over two decades, research has shown that some patients who meet the Ghent criteria for Marfan syndrome but lack *FBN1* pathogenic variants instead carry genetic variants in *TGFβR1* or *TGFβR2*. This discovery revealed significant clinical overlap between type 1 and type 2 Marfan syndrome (MFS1 and MFS2) and Loeys–Dietz syndrome (LDS), emphasizing the need for precise genetic and clinical differentiation [16].

The pleiotropic nature of TGFβ signaling has led to divergent findings regarding its function. As part of a larger signaling superfamily, it includes 30 different ligands, 7 distinct type I receptors, 5 type II receptors, and 8 Smad proteins [17,18]. Consequently, TGFβ signaling is tightly regulated at multiple levels, including extracellular modulation [19,20], transcriptional initiation by coactivators, corepressors, and transcriptional terminators [21], as well as various feedback mechanisms and cross-talk pathways that either terminate or redirect the intracellular signal [22,23].

This study provides important insights into the dysregulation of the TGFβ signaling pathway in Marfan syndrome (MFS). We observed overexpression of *FBN1* and underexpression of *TGFβ2*, *TGFβR1*, and *TGFβR2* in MFS patients, aligning with previous findings suggesting TGFβ signaling dysfunction in the context of MFS [24]. This altered gene expression may contribute to progressive aortic dilation and the high risk of aneurysms and aortic dissection observed in these patients [25].

Our findings regarding *FBN1* overexpression are consistent with the structural role of fibrillin in maintaining the integrity of the extracellular matrix (ECM) [26]. In MFS, pathogenic variants in *FBN1* impair fibrillin function, leading to the structural abnormalities observed in the aorta [27,28]. The increased *FBN1* expression we observed likely represents a compensatory response aimed at stabilizing the ECM to counteract these defects. However, despite this upregulation at the transcript level, the presence of pathogenic *FBN1* variants suggests that this compensation is functionally insufficient. In our cohort, we identified different types of FBN1 variants, including stop-gain mutations (*p.Gln1233Ter*, *p.Arg1541Ter*) and non-synonymous (*p.Gly753Val*, *p.Cys513Trp*, *p.Cys2378Tyr*) SNVs which likely impact fibrillin-1 function and contribute to ECM instability [29]. Previous studies have proposed that defective fibrillin incorporation into microfibrils may lead to increased gene expression as an attempt to restore ECM integrity, yet this overproduction does not necessarily result in functional protein assembly [15]. An apparent contradiction in our findings is the expected upregulation of *TGFβ2*, *TGFβR1*, and *TGFβR2* signaling in MFS, while *FBN1* is downregulated at the mRNA level. This paradox may stem from a failed feedback mechanism, where cells attempt to compensate for ECM dysfunction and increased mechanical stress by modulating gene expression [30]. This interplay between FBN1 deficiency and dysregulated TGFβ2 signaling may be a crucial factor in the vascular complications observed in MFS. While tissue inflammation and mechanical stress are known to activate TGFβ2 release from the ECM, the transcriptional suppression of key pathway components in our study suggests a disruption in intracellular regulatory mechanisms. This impairment may exacerbate the progression of aortic dilation, as TGFβ2 signaling plays a crucial role in ECM remodeling and vascular homeostasis [31].

Histopathological studies of aortas in MFS patients show significant inflammatory cellular infiltration, which can promote the release of pro-inflammatory cytokines that interfere with TGFβ signaling. The downregulation of *TGFβ2* observed in our study suggests that chronic inflammation in MFS may lead to a paradoxical suppression of key TGFβ signaling components, disrupting normal feedback loops [32]. The inflammatory infiltrate, including macrophages, T cells, and neutrophils, contributes to matrix degradation through the secretion of matrix metalloproteinases (MMPs), leading to further disruption of the ECM and abnormal TGFβ activation [33,34]. In particular, macrophages can release TGFβ2 trapped in the ECM, exacerbating aortic damage and initiating a cycle of inflammation and tissue degradation [35,36].

Previous studies have shown that mechanical stress from aortic dilation in MFS [37] can activate TGFβ signaling by releasing latent TGFβ2 bound to the extracellular matrix (ECM), which is then activated through mechanoreceptors [38]. However, in MFS, where the aortic wall is already compromised, this activation may be insufficient or dysregulated, failing to maintain structural integrity and instead contributing to aneurysm and dissection development. Consistent with these findings, research in murine models deficient in TGFβ2 receptors has demonstrated increased susceptibility to aortic complications, reinforcing the essential role of TGFβ in preserving aortic stability [39] and structural integrity [40,41,42]. These models also exhibit dysregulated ECM-related gene expression following TGFβ signaling disruption [43,44], further promoting aneurysm formation [45].

Our findings align with this evidence but reveal a key distinction; while *TGFβ2* signaling is often reported as upregulated at the extracellular level in various pathological conditions, we observed a significant downregulation of *TGFβ2* expression in the aortic tissue of MFS patients [46,47,48]. This suggests that mechanical stress, rather than consistently amplifying *TGFβ* signaling, may instead impair its intracellular regulation, disrupting normal mechanobiological responses and contributing to disease progression [49].

An important aspect of this study is the relationship between the expression levels of the evaluated mRNAs and the dilation of different aortic regions. In patients with Marfan syndrome (MFS), a positive correlation was observed between *FBN1* expression and the diameter of the sinotubular junction [50]. Similarly, a positive correlation was found between *TGFβ2* levels and the diameter of the ascending aorta, suggesting that *TGFβ2*-mediated signaling may play a role in the mechanisms leading to aortic dilation. It is possible that *TGFβ2* expression is altered as part of the cellular response to biomechanical stress on the aortic wall caused by dilation [51].

These findings align with studies showing the involvement of *TGFβ* in extracellular matrix remodeling and the fibrotic response in affected vascular tissues [52,53]. The increased *FBN1* levels observed in MFS may represent a compensatory response to ECM dysfunction, yet this compensation appears to be ineffective due to the presence of pathogenic variants [54].

Clinical manifestations of MFS vary depending on disease severity, with molecular alterations in key pathways influencing these manifestations. Elevated levels of active *TGFβ* are a prominent feature in this dysfunction, leading to reduced aortic wall distensibility and contributing to aneurysmal dilation [55]. This increases the risk of severe vascular complications.

The classic aortic aneurysm associated with Marfan syndrome, known as “annuloaortic ectasia”, involves dilation of the aortic ring and the sinuses of Valsalva. If not treated promptly, this condition can lead to aortic insufficiency, dissection, and congestive heart failure. As aortic dilation progresses, the risk of rupture and dissection increases. In MFS, the probability of dying from aortic disease is 200 times higher than in the general population [56].

In addition to aortic damage, patients may also have valve pathology, including a bicuspid aortic valve, which increases the risk of both aortic and cardiac valve dysfunction. Biomechanical studies have shown greater stiffness in the aortic tissue, which can contribute to early dilation and dissection. In adolescents and young adults, aortic valve regurgitation and aortic root disease are the most relevant cardiac pathologies, though mitral valve prolapse and regurgitation may also occur during aortic damage [57].

Given the complex pathophysiology at the cardiovascular level, it is crucial to analyze these intertwined damage mechanisms in detail to identify potential sites for therapeutic intervention. Our findings suggest that Marfan syndrome involves a maladaptive feedback loop, where compensatory mechanisms at the transcriptional level fail to counteract ECM dysfunction. This highlights the need for targeted therapies that can restore proper *TGFβ* signaling regulation and prevent progressive vascular damage.

## 4. Materials and Methods

### 4.1. Patient Population

A prospective comparative study was conducted between cases with MFS with aortic or aortic dissection and control subjects without MFS at the Ignacio Chavez National Institute of Cardiology between March 2014 and March 2023, including adults over 18 years of age of either sex who agreed to participate in the study. Twenty patients with MFS were included. In addition to the assessment consultation, conducted in the cardiology outpatient clinic, they were evaluated by a geneticist and were also re-evaluated by a rheumatology specialist expert in the subject, and the MFS was classified based on the 2010 Ghent criteria [17], which require more than two criteria for classification. These criteria are (1) a positive family history of Marfan syndrome (FH), (2) aortic dilatation and/or dissection (AoD), (3) ectopia lentis (EL), (4) a systemic score (SS) greater than 7/20, and/or (5) a mutation in the *FBN1* gene demonstrated by genetic testing. Fulfillment of any two criteria strongly suggests the syndrome. Echocardiogram, computed tomography or magnetic resonance imaging was performed to evaluate aortic enlargement or dissection, mitral and aortic valve prolapse, and tricuspid valve prolapse. Control subjects included (C) had aortic stenosis and Tri leaflet valve damage and did not have any syndromic or autoimmune pathology. The aortic dilatation that the controls presented is secondary to valvular dysfunction. The controls, like the patients with Marfan syndrome, required valve replacement surgery and aortic reconstruction. All patients and controls who presented aortic dilatation went to a medical–surgical session (Heart team) to standardize the requirements of the type of aortic surgery to be considered, according to the pathology of each patient. They were treated at the National Institute of Cardiology “Ignacio Chávez”. The criteria for exclusion were doubtful diagnosis and/or lack of agreement to sign the informed consent for the research study. The elimination criteria were considered as insufficient tissue sample taken at the time of surgery even when the patients met the inclusion criteria or inadequate conditions for obtaining the sample considering the requirements for the research process.

Mexican patients were defined as individuals with at least three generations born in Mexico.

Control subjects were 20 randomly selected subjects who underwent thoracic surgery for aortic stenosis and aortic dilatation.

### 4.2. Surgical Procedure

Patients were considered for surgery if they exhibited an aortic dissection or an aortic dilation of 4.5 cm or larger, confirmed by MRI or CT scans. An interdisciplinary team of cardiac specialists assessed the aortic and valvular conditions to choose the most appropriate surgical method. The procedures used included the Bentall and de Bono techniques, as well as the David-5 procedure, selected according to the specific needs and complexities of each case. The Bentall procedure involves replacing the aortic root and ascending aorta with a Dacron graft, to which both coronary arteries are attached, and one end is fitted with a valve prosthesis. On the other hand, the David Type 5 technique retains the patient’s own aortic valve and commissures, which are then reimplanted within the Dacron graft.

### 4.3. Blood Samples

Five milliliters of venous blood were collected in tubes with EDTA and without anticoagulant after 12 h of fasting. Plasma was separated by centrifugation for determination of the lipid profile.

### 4.4. Laboratory Analysis

Total cholesterol, triglycerides, and glucose were determined on a Hitachi 902 autoanalyzer (Böehringer Mannheim, Baden-Württemberg, Germany) using commercial enzymatic kits (Roche Diagnostics, Mannheim, Germany and WakoChemicals, Richmond, VA, USA). HDL-C was determined by a homogeneous enzymatic method (Roche Diagnostics, Mannheim, Germany).

### 4.5. Aortic Samples

Aortic tissues (2 mm) were collected from MFS patients and placed in sterile containers with RNA later (Quiagen, Hilden, Germany), which stabilizes and protects the tissues during processing and storage (−80 °C) prior to its processing.

### 4.6. Histology

A segment of 2 mm from the aortic tissues was processed according to conventional histological techniques and stained with hematoxylin and eosin (HE) staining, Weigert’s staining for elastic fibers, and Masson’s trichrome staining. Histological sections were acquired according to the method and equipment described by Zúñiga-Muñoz [58] and examined at 40× magnification.

### 4.7. Sample Pulverization

Frozen samples were pulverized using liquid nitrogen to maintain RNA integrity and prevent degradation. This process was carried out in a precooled mortar, pulverizing until a fine powder was obtained. The tissue was resuspended in a Tripure solution (a phenol and guanidine thiocyanate-based reagent) for total RNA extraction. The resuspension was performed following the manufacturer’s instructions for Tripure™ (Roche Molecular Biochemicals Welwyn Garden City, Hertfordshire, UK), ensuring proper tissue homogenization. The samples resuspended in Tripure were stored at −80 degrees Celsius until further processing. This storage ensures RNA stability and prevents its degradation.

### 4.8. mRNA Extraction and Quantification by RT-qPCR

mRNA was extracted using the TriPure Isolation Reagent technique (Roche Molecular Biochemicals, Welwyn Garden City, Hertfordshire, UK). RT-qPCR was then performed, which started with 1 μg of total RNA for cDNA synthesis according to the High-Capacity cDNA Reverse Transcription Kit (Applied Biosystems, Foster, CA, USA). mRNA was quantified using a Bio-Rad CFX96 Real-Time System (Bio-Rad, Hercules, CA, USA). Expression levels of human *FBN1* (Hs00171191_m1), *TGFβR1* (Hs00610320_m1), *TGFβR2* (Hs00234253_m1), *TGFβ2* (Hs00234244_m1) and *18s* rRNA (eukaryotic endogenous control 4319413E) (reference gene) were measured using a commercially available kit (TaqMan Gene Expression Assay, Applied Biosystems). Amplifications were performed by starting with a 10 min template denaturation step at 95 °C, followed by 40 cycles at 95 °C for 15 s and 60 °C for 1 min. All analyses were performed in duplicate. Data were expressed relative to each control value, and relative quantification was carried out using the formula 2^−ΔΔC^ [59].

### 4.9. Statistical Analysis

Data were analyzed using SPSS version 25 (SPSS Inc., Chicago, IL, USA). The risk factors were dichotomized in the form of presence or absence of clinical criteria. A descriptive analysis was conducted for all variables, with results expressed as mean ± standard deviation. The comparison between groups was made using the Student’s *t*-test for continuous variables. For variables that did not have a normal distribution, the Mann–Whitney U test or the Kruskal–Wallis test was used when we evaluated more groups.

### 4.10. Ethical Statement

Signed informed consent forms were obtained from each participant after fully explaining the purpose and nature of all procedures used in the study under the Declaration of Helsinki (10). The research was approved by the Ethical, Biosecurity, and Investigation Committees of the National Institute of Cardiology (registration number INCICh 23-1366).

### 4.11. Data Availability

Due to confidentiality agreements, the data underlying this study are not publicly available. Access to the data can be requested through c_huesca@yahoo.com following their confidentiality protocols.

## 5. Conclusions

Aortic dilatation in patients with Marfan syndrome is associated with key molecular changes, including decreased levels of *TGFβ2* and its receptors, *TGFβR1* and *TGFβR2*. On the other hand, aortic tissue damage is characterized by cystic necrosis and loss of smooth muscle cells, which may be the result of mechanical stress, proteolysis, inflammation and deregulation of oxidative stress, affecting various signaling pathways. Elevated levels of *FBN1* mRNA may reflect a compensatory response to this set of mechanisms involved in vascular damage. These findings regarding TGFβ and FBN1 mRNA in tissue support the continuation of molecular analysis in the serum of these patients, to better understand their clinical implications and potential therapeutic approaches.

## 6. Limitations

One limitation of this study is the small sample size, a common challenge in rare diseases. In MFS, cardiovascular complications are the primary life-threatening risk. As a national reference center, our institute treats most patients with connective tissue disorders. However, some cases are initially identified in ophthalmology clinics due to lens dislocation. These clinics often refer patients to our institute for a comprehensive evaluation. Similarly, pediatric centers transfer patients as they transition into adulthood, particularly when aortic complications arise. Given the incidence of MFS, the number of samples analyzed in this study is substantial. A more comprehensive understanding of the disease would require detailed genetic characterization beyond *FBN1*, including *TGFB2*, its receptors (*TGFBR1*, *TGFBR2*), and other genes involved in Marfan syndrome and related disorders. Expanding the cohort would necessitate multicenter studies with institutions in other countries that serve similar patient populations. This collaboration would allow for a deeper exploration of how specific genetic variants influence gene expression, signaling pathways, and disease progression. A key strength of this study is that no other national center processes as many samples, and all procedures are performed by expert surgeons.

## Figures and Tables

**Figure 1 ijms-26-03067-f001:**
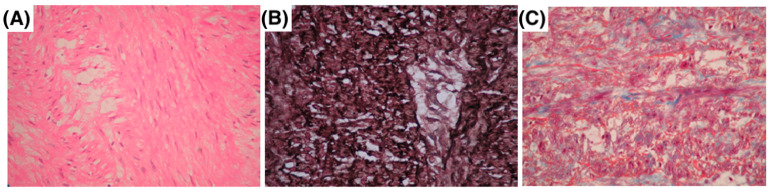
Histopathology of aortic aneurysms and dissections in three MFS patients. (**A**) A 28-year-old male with a giant ascending aorta aneurysm, showing rupture of elastic fibers in the adventitia with minimal inflammatory infiltrate hematoxylin–eosin. (**B**) A 36-year-old female with Stanford type A aortic dissection extending to the supra-aortic trunks, Weigert’s method. (**C**) A 33-year-old male with DeBakey type II aortic dissection, showing destruction of elastic fibers with cystic medial necrosis Masson’s trichrome staining.

**Figure 2 ijms-26-03067-f002:**
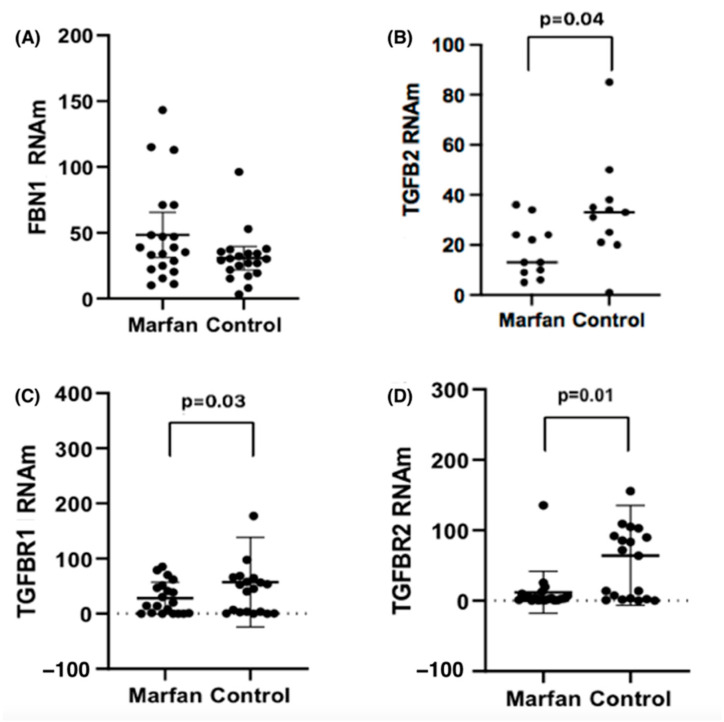
Expression levels of (**A**) *FBN1*, (**B**) *TGFβ2*, (**C**) *TGFβR1* and (**D**) *TGβR2* mRNA, normalized with 18 s mRNA, in the study groups.

**Table 1 ijms-26-03067-t001:** Demographic characteristics of patients with Marfan syndrome and controls.

	Total n = 40	MFS n = 20	Controls n = 20	*p*
Gender				
Women (n)	14	10	4	NS
Men (n)	26	10	16	NS
Age (years)	48 ± 19	32 ± 8	64 ± 13	0.0001
BMI (kg/m^2^)	25 ± 4	24 ± 4	26 ± 5	NS
Laboratories	Median (min–max)			
Hemoglobin (g/dL)	14 (9–28)	13 (9–18)	14 (9–28)	NS
Total Cholesterol (mg/dL)	258 (56–279)	150 (75–222)	179 (56–279)	0.03
HDL-C (mg/dL)	35 (8–68)	35 (19–54)	35 (8–68)	NS
LDL-C (mg/dL)	104 (40–202)	93 (40–132)	127 (40–202)	0.02
Triglycerides (mg/dL)	101 (41–371)	92 (41–371)	111 (80–396)	0.008
Glucose (mg/dL)	96 (54–209)	89 (54–209)	104 (81–271)	0.001
Serum Creatinine (mg/dL)	0.05 (0.6–5.0)	0.80 (0.6–1.7)	1.04 (0.7–5.0)	0.03
LVEF (%)	55 (15–71)	54 (18–66)	58 (15–71)	NS
Comorbidities				
Obesity n (%)	9 (23)	5 (25)	4 (20)	NS
SAH n (%)	19 (48)	5 (25)	14 (70)	0.0001
DMTII n (%)	10 (25)	4 (20)	6 (30)	NS
Smoking n (%)	2 (5)	0 (0)	2 (10)	NS
Hypothyroidism n (%)	2 (5)	0 (0)	2 (10)	NS
Aortic dilation n (%)	33 (83)	20 (100)	13 (65)	0.001
Atrial fibrillation n (%)	4 (10)	1 (5)	3 (15)	NS

BMI = body mass index, HDL-C = high-density lipoprotein cholesterol, LDL-C = low-density lipoprotein cholesterol, SAH = systemic arterial hypertension, DMTII = Diabetes Mellitus type II, LVEF = left ventricle ejection fraction. Non-normal variables are expressed as median (min–max) and analyzed with the Mann–Whitney U test, while normal variables are mean ± standard deviation and analyzed with the *t*-test. NS = no significance. *p* < 0.05.

**Table 2 ijms-26-03067-t002:** Characteristics of the total Ghent criteria with which the patients were classified and frequency of variables of the systemic score.

Patient	Age	Sex	Ghent Criteria						
			1	2	3	4		5	Total Ghent Criteria
			FH	EL	AD	Points SS	SS 7/20	FBN1	
1	32	F	+	−	+	15 (SWM, FF, PC; PN, DE, ER. HF, MS, SM, M, MVP)	+	TNP	3 (FH + AD + SS)
2	25	F	+	−	+	14 (SWM, FF, PC, HF, DE, MS, SM, AS/H)	+	+	4 (FH + AD + SS + FBN1)
3	36	F	+	−	+	13 (SWM, FF, PC, MS, ER, MVP)	+	TNP	3 (FH + AD + SS)
4	23	M	−	−	+	15 (SWM, PC, HF, RUS/LS, MS, M; MVP)	+	+	3 (AD + SS + FBN1)
5	35	F	−	+	+	7 (SWM, RUS/LS, MS, M, MVP)	+	TNP	3 (EL + AD + SS)
6	35	F	−	−	+	7 (FF, HF, RUS/LS, S M, MS, MVP)	+	TNP	2 (AD + SS)
7	52	F	−	−	+	7 (SWM, FF, HF, MS, SM, M, MVP)	+	TNP	2 (AD + SS)
8	51	M	−	−	+	8 (FF, MS, HF, ER, M, MS, DE, MVP)	+	TNP	2 (AD + SS)
9	35	F	+	+	+	10 (SWM, FF, PE, HF, DE, MS, MS)	+	TNP	4 (FH + EL + AD + SS)
10	33	M	−	−	+	4 (DE, MS, PC)	+	TNP	2 (AD + SS)
11	33	F	−	−	+	5 (PE, FF, HF, DE)	−	+	2 (AD + FBN1)
12	21	F	+	−	+	5 (PE, FF, DE; ER, MS)	−	+	3 (FH + AD + FBN1)
13	28	M	−	+	+	12 (SWM, FF, PE, HF, DE, MS, RUS/LS, ER, MS)	+	TNP	3 (EL + AD + SS)
14	37	M	+	+	+	9 (SWM, PE, HF, DEMS, SM)	+	+	5 (FH + EL + AD + SS + FBN1)
15	32	M	+	+	+	10 (SWM, FF, HF, DE, RUS/LS, MS, SM)	+	TNP	4 (FH + EL + AD + SS)
16	32	M	+	−	+	14 (SWM, FF, PE, HF, DE, RUS/LS, MS, ER, MS, M)	+	TNP	3 (FH + AD + SS)
17	30	M	−	−	+	11 (SWM, FF, PE, DE, ER; MS, M, MVP)	+	TNP	2 (AD + SS)
18	33	M	+	+	+	17 (SWM, FF, PE, HF, PN, DE, MD, AS/H, ER, MS, M, MVP)	+	TNP	4 (FH + EL + AD + SS)
19	22	M	+	+	+	9 (SWM, FF, PC, HF, MS, M, MVP)	+	TNP	4 (FH + EL + AD + SS)
20	51	F	−	−	+	7 (SWM, FF, MS, SM, MVP)	+	TNP	2 (AD + SS)
Frequency of systemic score items in patients with Marfan syndrome.									
							Total 20		
Stretch marks n (%)							19 (95)		
Facial features n (%)							17 (85)		
Cavus foot n (%)							14 (70)		
Scoliosis n (%)							13 (65)		
Reduction in the upper/lower segment n (%)							11 (55)		
Valve prolapse n (%)							11 (55)		
Dural ectasia n (%)							10 (50)		
Myopia n (%)							10 (50)		
Walker Murdock sign n (%)							9 (45)		
Pectum excavatum n (%)							7 (35)		
Pectum carinatum n (%)							6 (30)		
Steinberg sign n (%)							5 (25)		
Reduced elbow extension n (%)							3 (15)		
Pneumothorax n (%)							2 (10)		
Acetabular protrusion n (%)							0 (0)		

F = female, M = male, FH = family history, EL = ectopia lentis, AD = aortic dilatation, SS = systemic score, TNP = test not performed, SWM = Steinberg–Walker–Murdock sign, FF = facial feature, PC = pectum carinatum, PN = pneumothorax, DE = dural ecstasy, SM = stretch marks, M = myopia, MVP = mitral valve prolapse, HF = hollow foot, MS = mild scoliosis, ER = elbow reduction, R US/LS = reduction in the upper segment over the lower one, PE = pectum excavatum, AS > H = measurement of the arm span greater than the height.

**Table 3 ijms-26-03067-t003:** Type of surgery and type of surgical condition between cases and controls.

	Total	MFS	Controls	*p*
Type of surgery	n = 40	n = 20	n = 20	
Elective n (%)	31 (77.5)	16 (80)	15 (75)	NS
Urgent n (%)	9 (22.5)	4 (20)	5 (25)	NS
CSEC (minutes)	184 (122–440)	182 (122–352)	160 (94–440)	NS
Clamping time (minutes)	136 (66–254)	152 (110–224)	131 (66–254)	0.03
Postoperative Survival Duration (days)	19 (0–116)	−	16 (0–100)	NS
Without aortic dilation or dissection n (%)	7 (17.5)	1 (5)	6 (30)	NS
Aortic dilation n (%)	26 (65)	17 (85)	9 (45)	0.01
Aortic dissection n (%)	7 (17.5)	2 (10)	5 (25)	NS
Survivors n (%)	36 (90)	20 (100)	16 (80)	NS
They stopped coming to the institution	9 (22.5)	7 (35)	2 (10)	NS
Transfer to another institution n (%)	1 (2.5)	0 (0)	1 (5)	NS
Death n (%)	4 (10)	0 (0)	4 (20)	0.05

MFS = Marfan Syndrome, CSEC = cardiac surgery with extracorporeal circulation. Non-normal variables are expressed as median (min–max) and analyzed with the Mann–Whitney U test. NS = no significance. *p* < 0.05.

**Table 4 ijms-26-03067-t004:** Average aortic diameters, type of surgery and procedures performed in patients with aortic dilatation and MFS.

No	Sex	Age	CEC	ACT	ARD	SVD	STJ	TAscA	TS	D	Surgery
1	F	32	No	No	22	44	26	29	E	No	Aortic Replacement + Revascularization of Visceral Trunks
2	F	25	184	160	23	51	23	28	E	No	Post-Asc Ao Tube Abdominal Aorta Replacement + Mesenteric and Renal Trunks
3	F	36	272	197	22	40	34	28	E	No	Bentall + Replacement of Arch with 3 Trunks + Packing
4	M	23	149	113	43	91	95	30	E	No	Bentall + CVM INC 30
5	F	35	209	154	41	80	82	93	E	No	Bentall + REVAS HVSI-CD
6	F	35	187	149	44	50	45	53	E	No	Bentall
7	F	42	142	110	32	46	34	32	E	No	Bentall + Packing
8	M	51	192	145	30	75	60	68	E	No	Bentall + Tricuspid Valve Repair INC 28 + Packing
9	F	35	22	200	62	60	70	40	E	No	Bentall + CVM INC 34 + Tricuspid Valve Repair INC 28 + Left Atrial Redx + Arch Wrapping with Dacron Tube
10	M	33	151	122	38	44	36	36	E	No	Bentall
11	F	33	162	118	27	48	58	67	E	No	Bentall
12	F	21	210	115	38	57	72	54	U	yes	Bentall
13	M	28	184	159	36	94	100	107	E	No	Bentall + CVM INC 30 + REVAS HVSI-CD
14	M	37	179	150	37	44	45	38	E	No	Aortic Valve Resuspension (David Type)
15	M	32	186	136	46	77	77	76	E	No	Bentall
16	M	32	352	205	34	56	50	56	U	yes	Aortic Valve Replacement Bentall and Bono
17	M	30	290	224	35	82	37	37	U	yes	Aortic Valve Replacement Bentall and Bono
18	M	33	243	194	43	84	32	29	U	No	Aortic Valve Replacement Bentall and Bono
19	M	22	230	117	39	50	34	30	E	No	David
20	F	53	440	164	26	38	43	20	E	No	Bentall + Revascularization of Supra-Aortic Trunks + Elephant Trunk + Packing

CEC = extracorporeal circulation, ACT = aortic clamping time, ARD = aortic ring diameter, SVD = sinus of valsalva diameter, STJ = sinotubular junction, TAscA = tubular ascending aorta, TS = type of surgery, E = elective, U = urgent, D = dissection, F = female, M = male.

**Table 5 ijms-26-03067-t005:** Comparative analysis of aortic diameters in patients with Marfan syndrome and control group.

Diameters	Marfan	Control	*p*
Aortic root	37.55 ± 7.5	30.8 ± 6.1	0.004
Sinus of Valsalva	61.3 ± 20.9	38.4 ± 5.2	0.0001
Sinotubular junction	57.1 ± 21.4	38.2 ± 9.9	0.0001
Ascending aorta	58.8 ± 24.3	47.4 ± 15.5	0.08

Data are expressed as mean ± standard deviation.

**Table 6 ijms-26-03067-t006:** *FBN1*, *TGFβR1*, *TGFβR2* and *TGFβ2* levels between patients with Marfan syndrome and controls.

	With Dilatation Aneurysm			With Dissection		
mRNA	Controls n = 8	MFS n = 16	p1	Controls n = 3	MFS n = 4	p2
*FBN1*	33.04 (18.5–98.2)	34.5 (11–143.2)	NS	37.2 (25.1–79)	79 (48.1–110.1)	0.03
*TGFβR1*	60.6 (0.57–177.2)	35.3 (0.17–47.5)	NS	52.5 (6.8–65.8)	70.6 (30.1–111.1)	NS
*TGFβR2*	96.3 (0.05–282.02)	4 (0.40–41.3)	0.05	83.5 (71.8–135.9)	5.4 (3.6–7.2)	NS
*TGFβ2*	31.20 (1.12–35)	22.30 (5.60–36.86)	NS	44.29 (25.85–85.36)	12.17 (6.54–24.70)	0.02

MFS = Marfan syndrome; p1, statistical significance of comparative group that had dilatation, MFS vs. controls; p2, statistical significance of group with dissection, MFS vs. controls. Data are expressed as median (min–max) and analyzed with the Mann–Whitney U test. NS: no significance. *p* < 0.05.

**Table 7 ijms-26-03067-t007:** Correlation between mRNA expression levels and different aortic segments in Marfan patients and controls.

Aortic Segment Site	*FBN1*		*TGFβR1*		*TGFβR2*		*TGFβ2*	
Marfan	r	*p*	r	*p*	r	*p*	r	*p*
Aortic root	0.02	0.92	0.02	0.92	0.11	0.64	−0.03	0.87
Sinus of Valsalva	0.39	0.09	0.34	0.14	0.18	0.44	0.02	0.95
Sinotubular junction	0.42	0.07	0.22	0.35	0.02	0.90	0.44	0.05
Ascending aorta	0.07	0.74	0.42	0.07	0.14	0.55	0.53	0.01
Control	r	*p*	r	*p*	r	*p*	r	*p*
Aortic root	−0.20	0.52	0.21	0.91	0.33	0.28	0.62	0.004
Sinus of Valsalva	0.36	1.0	0.21	0.50	0.38	0.22	−0.46	0.039
Sinotubular junction	0.22	0.48	0.64	0.02	0.01	0.96	0.03	0.85
Ascending aorta	0.01	0.95	0.81	0.001	0.16	1.0	0.008	0.92

r: Pearson correlation coefficient; *p*: *p*-value.

**Table 8 ijms-26-03067-t008:** Differential expression of mRNA genes in Marfan syndrome patients with and without identified *FBN1* variants.

mRNA	Controls n = 20	MFS-Variants n = 5	MFS-TNP n = 15	p1	p2	p3
*FBN1*	29.53 (3.20–96.20)	33.86 (19–112.8)	29.08 (1.43–143.2)	0.27	0.75	0.45
*TGFβR1*	48.82 (0.10–347.5)	1.12 (0.25–61.79)	17.30 (0.7–78.79)	0.22	0.27	0.71
*TGFβR2*	67.86 (0.05–282.02)	2.03 (0.03–3.61)	4.03 (0.34–25.62)	0.02	0.01	0.16
*TGFβ2*	33.68 (1.12–85.36)	9.63 (5.60–13.67)	22.30 (6.54–36.86)	0.07	0.08	0.23

TNP = test not performed, p1 = Control vs. MFS-Variants, p2 = Control vs. MFS-TNP, p3 = MFS-Variants vs. MFS-TNP. Data are expressed as median (min–max) and analyzed with the Mann–Whitney U test. *p* < 0.05.

## Data Availability

Due to confidentiality agreements, the data underlying this study are not publicly available. Access to the data can be requested through c_huesca@yahoo.com following their confidentiality protocols.

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
