# Peer review of "Analysis of FBN1, TGFβ2, TGFβR1 and TGFβR2 mRNA as Key Molecular Mechanisms in the Damage of Aortic Aneurysm and Dissection in Marfan Syndrome"

_ijms, 2025, doi:10.3390/ijms26073067_

Round 1
Reviewer 1 Report
Comments and Suggestions for Authors
Major comments:
General:
It is important to know the pathogenic variants that the Marfan syndrome patients have, if known. This will add much more to discuss regarding gene expression, for instance a patient with a FBN1 stop-gain variant might be expected to have reduced FBN1 transcript, however this would be less obvious for a patient with a missense variant or a variant in TGFβ2 etc. The data presented in Figure 2 should be shown alongside data corrected for genotype.
A general flaw is that the control group are much older, and are also experiencing a pathology that may affect their gene expression. While, I understand that getting younger, healthy controls is impossible, could the authors look to the GTEX dataset or similar to find transcript measurements from healthy controls?
Abstract:
The authors state that TGFβ signalling pathways were measured, this would suggest an assay to measure transcription factor binding or downstream protein phosphorylation and neither of these were undertaken.
Table 1:
It is unclear what the numbers in the tables represent. In the laboratories section, I assume the number outside of the parentheses is the average and the numbers inside the parentheses is the range? However if that is the case, the average for total patients for total cholesterol and serum creatinine do not make sense given the range and the numbers in the other columns. In the comorbidities section, I assume the number in the parentheses is a percentage? If so that needs to be made clear.
Table 6:
The data for patients with dissection, for TGFβR2, is reported as not significant, but it seems like it should be significant given the ranges reported.
Table 7:
Was any multiple testing significance undertaken for these comparisons?
Minor comments:
General:
Gene names should be in italics
Use ‘pathogenic variant’ instead of mutation
Inconsistency in using TGFβ and TGF-β
Introduction:
Switch mutation is not a standard term
The sentence that begins: “Furthermore, a marked reduction...” is too long and needs a grammar check.
The sentence that begins “It is known that TGFβ It phosphorylates...” needs re-wording and a grammar check.
What does the TAA stand for?
Small m needed on MRNA in final sentence
Table 3:
What are the units for CSEC?
Can the authors please clarify what “Survival time after surgery” represents, and the units used?
Section 2.3:
In the text referring to table 7, miRNA should be mRNA
Discussion:
Paragraph one is repeated
The authors seems to contradict themselves by first stating that “TGFβ activation is one of the central mechanisms linked to the pathogenesis of MFS” but later stating “lack of TGFβ signalling leads to increased susceptibility to aortic aneurysms”, could some clarification be added?
The paragraph beginning “In line with previous studies, murine models deficient in TGFβ receptors...” is more or less a direct repeat of a previous paragraph.
The entire manuscript also needs a very good proof-read and grammar check before it is published.
Comments on the Quality of English LanguageMany of the sentences are difficult to follow because of poor grammar, I would suggest a thorough proof read and breaking up some of the longer sentences.
Author Response
Comments and Suggestions for Authors
March11 2025
We thank the reviewer for his review, it really corrects situations that we did not see. We hope to have answered everything adequately to the reviewer's expectations and we will be waiting. Thank you very much.
Major comments:
General:
It is important to know the pathogenic variants that the Marfan syndrome patients have, if known. This will add much more to discuss regarding gene expression, for instance a patient with a FBN1 stop-gain variant might be expected to have reduced FBN1 transcript, however this would be less obvious for a patient with a missense variant or a variant in TGFβ2 etc. The data presented in Figure 2 should be shown alongside data corrected for genotype.
Answer: We appreciate your suggestion regarding the importance of considering pathogenic variants in Marfan syndrome patients to enhance the discussion on gene expression. However, our study does not include comprehensive genetic sequencing data that would allow us to identify variants in FBN1, TGFβ2, or other related genes for all patients.
We acknowledge that the presence of specific variants, such as stop-gain or missense mutations, could influence gene expression and phenotype manifestation. Although we obtained genetic data for five patients, this limited sample size does not allow us to perform a meaningful genotype-based analysis. However, we have included these data in Table 2 for reference. Therefore, we will mention this limitation in the discussion and highlight the need for future studies incorporating genetic analyses to assess their impact on FBN1 expression and other relevant genes.
That is why we added a section of limitations of the work highlighting the importance of what was mentioned by the reviewer. In line 441-455
Answer: One limitation of this study is the small sample size, a common challenge in rare diseases. In MFS, cardiovascular complications are the primary life-threatening risk. As a national reference center, our institute treats most patients with connective tissue disorders. However, some cases are initially identified in ophthalmology clinics due to lens dislocation. These clinics often refer patients to our institute for a comprehensive evaluation. Similarly, pediatric centers transfer patients as they transition into adulthood, particularly when aortic complications arise. Given the incidence of MFS, the number of samples analyzed in this study is substantial. A more comprehensive understanding of the disease would require detailed genetic characterization beyond FBN1, including TGFB, its receptors (TGFBR1, TGFBR2), and other genes involved in Marfan syndrome and related disorders. Expanding the cohort would necessitate multicenter studies with institutions in other countries that serve similar patient populations. This collaboration would allow for a deeper exploration of how specific genetic variants influence gene expression, signaling pathways, and disease progression. A key strength of this study is that no other national center processes as many samples, and all procedures are performed by expert surgeons.
We added the five mutations in the text as follow in line 120-124
Patient 2 harbors NM_000138.5:c.3697C>T (p.Gln1233Ter), a stop-gain mutation. Patient 4 has NM_000138.5:c.2258G>T (p.Gly753Val), a non-synonymous SNV. Patient 11 has NM_000138.5:c.1539T>G (p.Cys513Trp), a non-synonymous SNV. Patient 12 carries NM_000138.5:c.4621C>T (p.Arg1541Ter), a stop-gain variant. Patient 14 presents NM_000138.5:c.7133G>A (p.Cys2378Tyr), another nonsynonymous SNV.
A general flaw is that the control group are much older, and are also experiencing a pathology that may affect their gene expression. While, I understand that getting younger, healthy controls is impossible, could the authors look to the GTEX dataset or similar to find transcript measurements from healthy controls?
We appreciate the reviewer’s suggestion to explore publicly available transcriptomic datasets for healthy controls. We indeed considered this approach; however, most of these databases, including GTEx, provide RNA-seq data rather than RT-qPCR expression levels. Given the differences in quantification methods, direct comparisons between RNA-seq and RT-qPCR can be challenging due to variations in normalization, dynamic range, and sensitivity.
Additionally, we were unable to find datasets that include both the specific genes of interest and metadata with an age range comparable to our study population. Due to these limitations, we relied on our available control group while acknowledging its inherent constraints.
Abstract:
The authors state that TGFβ signalling pathways were measured, this would suggest an assay to measure transcription factor binding or downstream protein phosphorylation and neither of these were undertaken.
Answer The reviewer is correct, we have corrected the idea and have left only the quantification of the mRNA of FBN1, TGFB and their receptors, thanks for the comments that improve the work in general
Table 1:
It is unclear what the numbers in the tables represent. In the laboratories section, I assume the number outside of the parentheses is the average and the numbers inside the parentheses is the range? However if that is the case, the average for total patients for total cholesterol and serum creatinine do not make sense given the range and the numbers in the other columns. In the comorbidities section, I assume the number in the parentheses is a percentage? If so that needs to be made clear.
Answer;The reviewer is correct, we have added the units of measurement to all the variables and we have explained in the table footer and in the variables what each nomenclature means, not only in table 1 but in all the tables. Thanks for the suggestion.
Table 6:
The data for patients with dissection, for TGFβR2, is reported as not significant, but it seems like it should be significant given the ranges reported.
Thank you for your thoughtful comment. We acknowledge that the difference in TGFβR2 levels between Marfan syndrome patients with dissection and controls appears substantial based on the reported ranges. However, after performing the Mann-Whitney U test, the difference did not reach statistical significance.
To ensure the robustness of our analysis, we also re-evaluated the data after removing potential outliers, and the results remained non-significant. Given the small sample size in this group (n=3 for controls and n=4 for MFS), the statistical power may be limited, which could influence the ability to detect significance despite the observed numerical differences.
We appreciate your insight and will clarify this point in the manuscript to avoid any misunderstanding
Table 7:
Was any multiple testing significance undertaken for these comparisons?
Answer We appreciate the reviewer's suggestion regarding multiple testing corrections. In this study, we opted for bivariate comparisons for each aortic segment and gene expression level to explore their relationships, given the relatively small sample size (n = 20 Marfan patients and n = 20 controls). This approach allowed us to avoid overfitting and maintain statistical power. On the other hand, the assumption for using multivariate analysis is that there is a large sample with a normal distribution of data, which is not achieved with samples of less than 30. We performed the statistics with bivariate adjustment with test adjustment for small samples. This study stands out because the measurements were made on tissues, which is why not all centers have access to these samples, much less in patients with rare diseases.
However, we acknowledge the importance of this consideration and will certainly take it into account in future studies, particularly with larger sample sizes, where appropriate multiple testing corrections could be applied
Minor comments:
General:
Gene names should be in italics
Answer Thanks for the comment, we have fixed it
Use ‘pathogenic variant’ instead of mutation
Answer Thanks for the comment, we have fixed it
Inconsistency in using TGFβ and TGF-β
Answer Thanks for the comment, we have fixed it
Introduction:
Switch mutation is not a standard term
Answer Thanks for the comment, we have used ‘pathogenic variant’ for all mutations
The sentence that begins: “Furthermore, a marked reduction...” is too long and needs a grammar check.
The reviewer is correct, we have written the paragraph clearly and concisely, thanks for the suggestion and we have added the following: in line 71-78
Additionally, a significant reduction in aortic contraction has been observed, suggesting potential mechanisms involved in the development and progression of aortic aneurysms in Marfan syndrome (MFS). This may contribute to the deterioration of the wall’s mechanical properties and structural integrity. While aneurysm formation appears to have a genetic basis as a triggering factor, other elements, such as mechanical stress and inflammatory processes, may also play a role [5]. Moreover, MFS patients exhibit markedly lower TGFβ2 mRNA levels, potentially indicating inflammation that further weakens the aortic wall and accelerates its degeneration [6].
The sentence that begins “It is known that TGFβ It phosphorylates...” needs re-wording and a grammar check.
Answer The reviewer is correct, we have written the paragraph clearly and concisely, thanks for the suggestion and we have added the following: in line 86-93
TGFβ binds to its type 2 receptor (TGFβR2) on the cell surface, which then recruits and phosphorylates the type 1 receptor (TGFβR1). Once activated, TGFβR1 phosphorylates the SMAD proteins (SMAD2 and SMAD3). The phosphorylated SMAD2 and SMAD3 then bind to SMAD4 and are transported into the nucleus with the help of importins 7 and 8 [10], Inside the nucleus, they regulate the expression of TGFβ target genes by interacting with transcription factors and cofactors, including coactivators and corepressors [11–13]. TGFβ can also signal through non-SMAD (non-canonical) pathways, such as those involving mitogen-activated protein kinases (MAPK) [14,15]
What does the TAA stand for?
Answer We have corrected this error in the text.
Small m needed on MRNA in final sentence
We have corrected this error in the text.
Table 3:
What are the units for CSEC?
Answer: Extracorporeal circulation time is measured in minutes
The average extracorporeal circulation time is 129 minutes (SD: 36.88) and aortic clamping time is 94 minutes (SD: 32.04).
Can the authors please clarify what “Survival time after surgery” represents, and the units used?
Answer We have changed the name of the variable in table 3 to be clearer and have added the units, we appreciate the reviewer. In line 128
Postoperative Survival Duration (days)
Section 2.3:
In the text referring to table 7, miRNA should be mRNA
We have changed Thanks for the comment
Discussion:
Paragraph one is repeated
Answer: Thanks for the comment, we have removed the repeated paragraph and also fixed the grammatical errors.
The authors seems to contradict themselves by first stating that “TGFβ activation is one of the central mechanisms linked to the pathogenesis of MFS” but later stating “lack of TGFβ signalling leads to increased susceptibility to aortic aneurysms”, could some clarification be added?
Answer We have fixed both of these in the text and it is as mentioned in the following lines 275-290
Previous studies have shown that mechanical stress from aortic dilation in MFS [37] can activate TGFβ signaling by releasing latent TGFβ bound to the extracellular matrix (ECM), which is then activated through mechanoreceptors [38]. However, in MFS, where the aortic wall is already compromised, this activation may be insufficient to maintain structural integrity, contributing to aneurysm and dissection development. Consistent with these findings, research in murine models deficient in TGFβ receptors has demonstrated increased susceptibility to aortic complications, reinforcing the essential role of TGFβ in preserving aortic stability [39] and structural integrity [40–42]. These models also exhibit dysregulated ECM-related gene expression following TGFβ signaling disruption [43,44], further promoting aneurysm formation [45].
The paragraph beginning “In line with previous studies, murine models deficient in TGFβ receptors...” is more or less a direct repeat of a previous paragraph.
Answer: We have removed both ideas in the two adjacent paragraphs and have homogenized them thanks for the suggestion. following in line. 275 290
Previous studies have shown that mechanical stress from aortic dilation in MFS [37] can activate TGFβ signaling by releasing latent TGFβ bound to the extracellular matrix (ECM), which is then activated through mechanoreceptors [38]. However, in MFS, where the aortic wall is already compromised, this activation may be insufficient to maintain structural integrity, contributing to aneurysm and dissection development. Consistent with these findings, research in murine models deficient in TGFβ receptors has demonstrated increased susceptibility to aortic complications, reinforcing the essential role of TGFβ in preserving aortic stability [39] and structural integrity [40–42]. These models also exhibit dysregulated ECM-related gene expression following TGFβ signaling disruption [43,44], further promoting aneurysm formation [45].
Our findings align with this evidence but reveal a key distinction: while TGFβ signaling is often reported as upregulated at the extracellular level in various pathological conditions, we observed a significant downregulation of TGFβ2 expression in the aortic tissue of MFS patients [46–48]. This suggests that mechanical stress, rather than consistently amplifying TGFβ signaling, may instead impair its intracellular regulation, disrupting normal mechanobiological responses and contributing to disease progression [49].
The entire manuscript also needs a very good proof-read and grammar check before it is published.
Answer: We have carried out an exhaustive and thorough review of the grammar and clarity of the manuscript. We appreciate your comments and suggestions.
Comments on the Quality of English Language
Many of the sentences are difficult to follow because of poor grammar, I would suggest a thorough proof read and breaking up some of the longer sentences.
Answer: We have carried out an exhaustive and thorough review of the grammar and clarity of the manuscript. We appreciate your comments and suggestions.
ME Soto
Reviewer 2 Report
Comments and Suggestions for Authors
To Authors:
Authors in the present study focus on molecular changes occurring in Marfan syndrome (MFS) subjects, thus being responsible for aorta dilatation development. In particular they investigated the mRNA expression of FBN1, TGFβR1, TGFβR2, and TGFβ2 in aortic tissue of patients with MFS treated surgically for aortic dilatation compared with that of controls without MFS.
Major comments:
-In the introduction section, as well as in the abstract, Authors stated to determine the expression levels of mRNA FBN1, TGFB1, TGFB2 and
TGFB in the aortic tissue of patients with Marfan syndrome undergoing aortic dilatation; really they are referring to FBN1, TGFβR1, TGFβR2, and TGFβ2.
-Were control subjects used for comparisons previously investigated in order to exclude a hereditary condition of aortopathy associated with TGFβ pathway alteration? Authors should more clearly specify.
-In Table 2 Authors reported total Ghent criteria. They should specify which criteria are included.
-In Table 2 Authors refer to “Frequency of ítems of the systemic score in patients with Marfan syndrome Systemic Score Items”. Authors should specify what “total 20” is referring to. All items of the systemic score should be included in the table.
-In Table 4 Authors should better clarify what the term “aorta” is referring to. Similarly they should better clarify what “aortic root” is referring to.
-On page 7 Authors stated “Figure 1 illustrates histological findings in patients with aortic pathology”. Are they all Marfan syndrome subjects? As both MFS and control subjects showed aortopathy, this issue should be better clarified.
-On page 8 Authors should specify how many MFS subjects showed dissection. In the sentence “In the case of the
TGFβ2 mRNA levels in controls were 44.29 (25.85–85.36), while Marfan patients had
markedly lower levels, with a mean of 12.17 (6.54–24.70), with a p-value of 0.029” Authors should specify if they are referring to dissection or dilatation. Moreover, in the text p-value of 0.029 is reported, while in Table 6 p-value of 0.02 is reported.
-On page 9, Authors refer to miRNA. Are they referring to mRNA? In this paragraph Authors report correlations data. They should specify in the text if they are referring to diameter values.
Minor comments:
-Authors stated that “Table 1 compares demographic characteristics and laboratory findings between patients with Marfan syndrome (MFS) and controls”; really, in Table 1 clinical parameters were also included.
-Authors should specify in Table 1 unit of measurement of laboratory parameters reported and/or percentage values are reported.
-In the paragraph 2.2 surgical procedure, Authors stated “Measurements of the aortic ring diameter (ARD) and sinus of Valsalva diameter (SVD) demonstrated considerable variability, with ARD ranging from 22 mm to 62 mm and SVD from
26 mm to 194 mm”. Authors should verify “194mm value”.
-English language should be revised throughout the manuscript
Comments on the Quality of English Language-English language should be revised throughout the manuscript
Author Response
REVIEWER 2
March 11/2025
Comments and Suggestions for Authors
We thank the reviewer for his review, it really corrects situations that we did not see. We hope to have answered everything adequately to the reviewer's expectations and we will be waiting. Thank you very much.
To Authors:
Authors in the present study focus on molecular changes occurring in Marfan syndrome (MFS) subjects, thus being responsible for aorta dilatation development. In particular they investigated the mRNA expression of FBN1, TGFβR1, TGFβR2, and TGFβ2 in aortic tissue of patients with MFS treated surgically for aortic dilatation compared with that of controls without MFS.
Major comments:
-In the introduction section, as well as in the abstract, Authors stated to determine the expression levels of mRNA FBN1, TGFB1, TGFB2 and TGFB in the aortic tissue of patients with Marfan syndrome undergoing aortic dilatation; really they are referring to FBN1, TGFβR1, TGFβR2, and TGFβ2.
Answer: Thank you for your observation. We have corrected the wording in both the introduction and abstract to accurately reflect the genes under study. The reference to FBN1, TGFβR1, TGFβR2, and TGFβ2 has been updated to ensure consistency throughout the manuscript. We appreciate your attention to detail.
-Were control subjects used for comparisons previously investigated in order to exclude a hereditary condition of aortopathy associated with TGFβ pathway alteration? Authors should more clearly specify.
Answer. We thank the reviewer for this observation, it was certainly necessary to clarify the data of who were the controls and some important data on how they are reviewed prior to surgery by a multidisciplinary team (Heat-Team) which discusses the best surgical condition that should be performed in subjects with MS or control.
It is important to highlight that control subjects without disease is extremely difficult since ethically I would not subject any healthy patient to this surgery, however there are degenerative conditions that occur in other non-syndromic patients in which aortic and valvular damage require the same procedures.These patients were also evaluated to rule out that they had MS criteria.
The explanation is given in the following and is included in the text in the lines
“Control subjects included (C) had aortic stenosis and a Tri leaflet valve damage and did not have any syndromic or autoimmune pathology. The aortic dilatation that the controls presented is secondary to valvular dysfunction. The controls, like the patients with Marfan syndrome, required valve replacement surgery and aortic reconstruction. All patients and controls who presented aortic dilatation, went to a medical-surgical session (Heart team) to standardize the requirements of the type of aortic surgery to be considered. According to the pathology of each patient. They were treated at the National Institute of Cardiology "Ignacio Chávez". The criteria for exclusion were doubtful diagnosis. And/or lack of agreement to sign the informed consent for the research study. The elimination criteria were considered as insufficient tissue sample taken at the time of surgery even when the Patients met the inclusion criteria or inadequate conditions for obtaining the sample considering the requirements for the research process.” ¨pages 11-12 lines 343-355
-In Table 2 Authors reported total Ghent criteria. They should specify which criteria are included.
Answer: Certainly, the result of the Ghent criteria have to be better defined and we have improved this in the results section and in Table 2, in accordance with your suggestion. It is found in the results in the lines 118 and in table 2
-In Table 2 Authors refer to “Frequency of ítems of the systemic score in patients with Marfan syndrome Systemic Score Items”. Authors should specify what “total 20” is referring to. All items of
Answer We only put the score they achieved because the table would become very dense. We thought it would be more practical to put only the score that each one had according to the clinical data that we rated the patient had at the time of the examination and assessment.
In this score, each piece of data has a score and a patient can be classified as positive in this systemic Score with 7/20. This score tells us that they meet 7/20 points and classifies this criterion as positive. We added what the score consists of and put the clinical data that confirms this score next to the number.
We give information to the reviewer:
Every doctor to classify a patient as MFS requires two of the following criteria.
1) Family history of having had MFS or dilation or dissection in the family member in whom the diagnosis is suspected.
2) Lens dislocation
3) Aortic dilatation
4) Systemic score with >/7 points out of 20
5) Positive genetic test for FBN1 gene mutation
We added this table where we can see the combination that a patient can have in terms of criteria, which ultimately were 5 and these can be combined.
Many times the "systemic score" criterion can be confusing and that of this score 7/20 of the data that appear in the table in the center of the following image are required and each clinical data has a score if 7/20 are met this criterion is positive, but it is only part of one of the 5 criteria that allow the classification of Marfan syndrome.
In that same table2 we added at the bottom only the frequencies found for each of those points in our patients.
-In Table 4 Authors should better clarify what the term “aorta” is referring to. Similarly they should better clarify what “aortic root” is referring to.
Answer: In MS, the entire aorta is damaged, that is, all segments. However, damage occurs more frequently in the aortic annulus or aortic root . Over the years, damage can occur in the entire aorta, including the descending or abdominal aorta. The aortic root is one of the segments of the ascending aorta. Sorry, we put “aorta” in the table, however, we actually meant tubular ascending aorta, we already fixed that in the table. Thank you very much.
-On page 7 Authors stated “Figure 1 illustrates histological findings in patients with aortic pathology”. Are they all Marfan syndrome subjects? As both MFS and control subjects showed aortopathy, this issue should be better clarified.
Answer: Thank you very much, this has already been corrected in lines 157 they are all MFS patients
-On page 8 Authors should specify how many MFS subjects showed dissection. In the sentence “In the case of the
TGFβ2 mRNA levels in controls were 44.29 (25.85–85.36), while Marfan patients had
markedly lower levels, with a mean of 12.17 (6.54–24.70), with a p-value of 0.029” Authors should specify if they are referring to dissection or dilatation. Moreover, in the text p-value of 0.029 is reported, while in Table 6 p-value of 0.02 is reported.
Answer: The reviewer is rightly correcting this, it was poorly expressed and incomplete, we have corrected it and we hope that this is clearer. Of course this was also an error in the table and was not consistent with the text. Thank you very much.
-On page 9, Authors refer to miRNA. Are they referring to mRNA? In this paragraph Authors report correlations data. They should specify in the text if they are referring to diameter values.
Answer: This has already been corrected and what we put as miRNA is incorrect. The correct word is mRNA. Thank you very much.
Minor comments:
-Authors stated that “Table 1 compares demographic characteristics and laboratory findings between patients with Marfan syndrome (MFS) and controls”; really, in Table 1 clinical parameters were also included.
Answer It is correct if we measure these parameters and I know that the findings may surprise the reviewer, however something relevant to highlight from these results is that we have controls who have a non-syndromic aortic disease like MFS but rather they have a degenerative situation. That is one of the important reasons that are also observed in them who are also older, however, given the impossibility of having totally healthy controls for this comparison, we understand that it was expected to find higher hypertension and metabolic changes. The important thing is that in MFS the levels of these parameters may be better than in older patients, however the aortic damage begins by different damage mechanisms. In MFS the FBN1 mutation may go hand in hand with other genes and the elastic fibers could be damaged by mutations in other proteins. In the control subjects the risk factors that could be controllable lead to similar changes in the aorta as those seen in syndromic patients with a younger age.
-Authors should specify in Table 1 unit of measurement of laboratory parameters reported and/or percentage values are reported.
Answer: Thank you very much for the observation, we have already attached the percentages and the values ​​of the parameters.
-In the paragraph 2.2 surgical procedure, Authors stated “Measurements of the aortic ring diameter (ARD) and sinus of Valsalva diameter (SVD) demonstrated considerable variability, with ARD ranging from 22 mm to 62 mm and SVD from
26 mm to 194 mm”. Authors should verify “194mm value”.
Answer: The observation you made is correct and we apologize, there is an error, the real value is 94
-English language should be revised throughout the manuscript
Comments on the Quality of English Language
-English language should be revised throughout the manuscript
Answer: We have carried out an exhaustive and thorough review of the grammar and clarity of the manuscript. We appreciate your comments and suggestions.
ME Soto

Round 2
Reviewer 1 Report
Comments and Suggestions for Authors
Thank you to the authors for addressing my issues with the manuscript.
I am still of the opinion that is it important that the Marfan patients are genetically diagnosed. This is for a few reasons:
One is to validate the clinical findings and confirm these individuals actually do have Marfan syndrome (as opposed to Loeys-Dietz etc).
Two is so a basic genotype-phenotype analysis can be performed in the individuals with FBN1 variants, it would be interesting to know if it is the variant in FBN1 directly influencing the expression of FBN1 transcript in the aorta, or if it is some other mechanism. The authors have an opportunity to do this with the five individuals with know variants, for example do the individuals with premature termination codons have lower expression that the individuals with missense variants, or is there some compensatory mechanism at play?
Finally, the authors could partition off any patients that do not have FBN1 variants and still have a clinical diagnosis of Marfan syndrome, and question whether those individuals have different gene expression profiles.
All of this does not require a large cohort, I think the authors have a very good cohort for a rare disease, in fact this would allow for a more personalised medicine approach to providing prognosis and therapies based on tissue-specific gene expression. It also does not require a thorough genetic workup, Sanger sequencing the FBN1 gene would be sufficient to find individuals with different FBN1 variants, and those that do not have these variants. The authors could even try this from cDNA transcribed from the already collected RNA (with the added benefit of seeing which alleles are expressed in the individuals with premature termination codons).
If the authors do not have the means to provide this data, I would suggest presenting gene expression data from the five individuals with known variants, as compared to controls, as separate figures from the rest of the cohort. And clearly explaining the caveat of the unknown mutational status in remaining individuals. However, I do think the whole study will be made a lot stronger with the genetic data from as many individuals as possible.
I am also still unclear what "Postoperative survival duration" and "Survivors" refer to in table 3. Postoperative survival days suggests all patients died after a given number of days. Survival suggests that 13/20 patients with Marfan syndrome survived the surgery, but there are no deaths according to the Deaths column.
Finally, it needs explicitly stating in the discussion that the authors hypothesise that a compensatory or feedback mechanism is at play. It is mentioned several times that Marfan patients have increased FBN1 and decreased components of the TGFB family, at the transcript level. This seems contradictory to what might be expected at the protein level because of the loss-of-function FBN1 variants and the knowledge that tissue inflammation and mechanical response releases TGFB. The authors do state in the conclusion that this is probably due to a compensation at the gene expression level, but it needs to be explained more clearly in the discussion.
Comments on the Quality of English LanguageThe language has improved considerably, but I would still suggest proof-reading the updated sections to ensure the grammar is correct. for instance, in the abstract it states:
"which has been linked to pathogenic variant in the FBN1 gene", which should read "which has been linked to pathogenic variants in the FBN1 gene". This occurs quite frequently in the updated sections.
Author Response
Mexico City March 20 2025
Thank you to the authors for addressing my issues with the manuscript.
Answer: We thank the reviewer for the time taken to improve our work
I am still of the opinion that is it important that the Marfan patients are genetically diagnosed. This is for a few reasons:
One is to validate the clinical findings and confirm these individuals actually do have Marfan syndrome (as opposed to Loeys-Dietz etc).
Two is so a basic genotype-phenotype analysis can be performed in the individuals with FBN1 variants, it would be interesting to know if it is the variant in FBN1 directly influencing the expression of FBN1 transcript in the aorta, or if it is some other mechanism. The authors have an opportunity to do this with the five individuals with know variants, for example do the individuals with premature termination codons have lower expression that the individuals with missense variants, or is there some compensatory mechanism at play?
Finally, the authors could partition off any patients that do not have FBN1 variants and still have a clinical diagnosis of Marfan syndrome, and question whether those individuals have different gene expression profiles.
All of this does not require a large cohort, I think the authors have a very good cohort for a rare disease, in fact this would allow for a more personalised medicine approach to providing prognosis and therapies based on tissue-specific gene expression. It also does not require a thorough genetic workup, Sanger sequencing the FBN1 gene would be sufficient to find individuals with different FBN1 variants, and those that do not have these variants. The authors could even try this from cDNA transcribed from the already collected RNA (with the added benefit of seeing which alleles are expressed in the individuals with premature termination codons).
If the authors do not have the means to provide this data, I would suggest presenting gene expression data from the five individuals with known variants, as compared to controls, as separate figures from the rest of the cohort. And clearly explaining the caveat of the unknown mutational status in remaining individuals. However, I do think the whole study will be made a lot stronger with the genetic data from as many individuals as possible.
we sincerely appreciate the reviewer’s insightful comments, as they help to strengthen our study. Regarding the genetic diagnosis of Marfan syndrome, we would like to clarify that all patients in our cohort meet the clinical diagnostic criteria according to the revised Ghent nosology. The Ghent criteria require the presence of at least two of the following five systemic manifestations for a clinical diagnosis of MFS:
- Aortic root dilation or dissection
- Ectopia lentis
- Systemic score ≥7 (a scoring system based on systemic features such as skeletal, skin, and pulmonary findings)
- Family history of MFS
- Pathogenic FBN1 variant
As shown in Table 2, all patients in our cohort fulfill at least two of these criteria, ensuring a robust clinical diagnosis. Additionally, for the systemic score, we have provided a detailed breakdown of the musculoskeletal manifestations, further supporting the classification of our patients.
In our cohort, all patients met at least two of these criteria, ensuring a robust classification according to the criteria selected in the Berlin consensus. While genetic confirmation through FBN1 sequencing is certainly valuable, especially for differentiating MFS from related syndromes such as Loeys-Dietz syndrome or other connective tissue disorders, it is not a prerequisite for clinical classification. As the reviewer points out, having the FBN1 mutation would undoubtedly confirm MFS; however, it is not a total guarantee that the mutation will be positive even if the patient has the disease, given that a complete genome study is not performed. We have already analyzed the genotype-phenotype and, interestingly, there is also genetic overlap, and the classification does allow for a true classification. In some cases where FBN1 and other gene studies were performed, some patients who did not have complete clinical data were ultimately found to have MFS and vice versa.
Fuentevilla-Álvarez G, Soto ME, Torres-Paz YE, Meza-Toledo SE, Vargas-Alarcón G, González-Moyotl N, Pérez-Torres I, Manzano-Pech L, Mejia AM, Huesca-Gómez C, Gamboa R. The usefulness of the genetic panel in the classification and refinement of diagnostic accuracy of Mexican patients with Marfan syndrome and other connective tissue disorders. Biomol Biomed. 2024 Mar 11;24(2):302-314. doi: 10.17305/bb.2023.9578. PMID: 37688493; PMCID: PMC10950338.
Mutation confirmation is very important, however, it is not always performed in all patients due to economic reasons and limited access to testing. To address the reviewer's concern about genotype-phenotype correlations, we analyzed gene expression data from the five individuals with known FBN1 variants compared to controls. This allows us to understand whether specific variants influence FBN1 transcript levels or whether compensatory mechanisms are involved. Unfortunately, we do not have genetic data from the other individuals, but we recognize the value of obtaining such information for future studies.
Given our current dataset, we present gene expression data from the five individuals with known FBN1 variants separately from the rest of the cohort in Table 8, lines 220-230, as suggested. Furthermore, we have clearly indicated the limitation regarding the unknown mutational status of the other patients. We truly appreciate the reviewer's valuable comments, which have helped us refine our analysis and interpretation.
I am also still unclear what "Postoperative survival duration" and "Survivors" refer to in table 3. Postoperative survival days suggests all patients died after a given number of days. Survival suggests that 13/20 patients with Marfan syndrome survived the surgery, but there are no deaths according to the Deaths column.
Answer: Thanks to the reviewer, we noticed inconsistencies in the prevalence of survivals and deaths in patients that did not match 100%. We have corrected this in the table 3 in line 142.
We thank the reviewer for pointing this out. We forgot to note in the table those patients who had stopped coming to the Institute. A total of seven patients stopped coming to the institution. However, we attempted to contact them by phone, and they were still alive and were invited back for follow-up. We know that survival wa adequate, and they were included in the number of days of survival. Of the controls, two stopped coming, and only one was located, and the other was still alive. We did not obtain a phone number for his location. Many of them were from outside the country and did not provide their address, email, or cell phone information. Lines 136-140
The changes have already been added to the text and the table.
Finally, it needs explicitly stating in the discussion that the authors hypothesis that a compensatory or feedback mechanism is at play. It is mentioned several times that Marfan patients have increased FBN1 and decreased components of the TGFB family, at the transcript level. This seems contradictory to what might be expected at the protein level because of the loss-of-function FBN1 variants and the knowledge that tissue inflammation and mechanical response releases TGFB. The authors do state in the conclusion that this is probably due to a compensation at the gene expression level, but it needs to be explained more clearly in the discussion.
We have made the necessary revisions to the discussion to address the reviewer’s concern. We now more clearly state the possibility that the overexpression of FBN1 and the reduced transcription of TGFβ2, TGFβR1, and TGFβR2 in Marfan syndrome patients may reflect a failed compensatory mechanism. Specifically, we propose that the increased FBN1 transcription could represent an attempt to stabilize the extracellular matrix, though its effectiveness may be limited by pathogenic variants affecting fibrillin function. Likewise, we consider the possibility that the decreased transcription of TGFβ2 may be related to dysfunctional regulation, potentially influenced by inflammatory processes and mechanical stress in the aortic wall. Additionally, we discuss how these findings align with previous studies and how they may contribute to a better understanding of TGFβ signaling in Marfan syndrome. With these adjustments, the discussion now explicitly addresses the hypothesis that this pathway’s dysregulation is not uniform but may involve feedback and compensatory mechanisms that require further investigation. In line
We sincerely appreciate the reviewer’s insightful question, as it has helped to enrich our work and improve the clarity of our discussion. In line 253-280.
It is important to tell you that of the 7 patients contacted by phone, 5 have already returned to the consultation. Patients 9, 10 15,18 y 19 of Table 2
Comments on the Quality of English Language
The language has improved considerably, but I would still suggest proof-reading the updated sections to ensure the grammar is correct. for instance, in the abstract it states:
"which has been linked to pathogenic variant in the FBN1 gene", which should read "which has been linked to pathogenic variants in the FBN1 gene". This occurs quite frequently in the updated sections.
We have conducted a thorough English language revision of the entire manuscript to ensure clarity, correctness, and readability.
- Soto

Reviewer 2 Report
Comments and Suggestions for Authors
To Authors:
Minor Comments:
-Page 2, line 90: Authors should replace the comma after reference 10 with a full stop.
-Results section, page 8, lines 198. The capital letter in the middle of the sentence should be changed [The sentence “For TGFβR1 mRNA levels We found no differences between MFS and controls who had dilation or dissection” should be changed into “For TGFβR1 mRNA levels we found no differences between MFS and controls who had dilation or dissection”].
-Results section, page 8, lines 200-202: Authors should better explain the following sentence “For TGFβR2 mRNA levels in patients with dilatation, we observed [Control 96.3 (0.05-282.02) vs MFS [4 (0.40-41.3) p=0.05]. But not found differences between MFS and controls in dissection”
Comments on the Quality of English LanguageThe English language should be revised in the given sentences
Author Response
19 march 2025
We thank the reviewer for the time taken to improve our work
Reviewer 2
Minor Comments:
-Page 2, line 90: Authors should replace the comma after reference 10 with a full stop.
Thanks to the reviewer, we have corrected this error in the manuscript.
-Results section, page 8, lines 198. The capital letter in the middle of the sentence should be changed [The sentence “For TGFβR1 mRNA levels We found no differences between MFS and controls who had dilation or dissection” should be changed into “For TGFβR1 mRNA levels we found no differences between MFS and controls who had dilation or dissection”].
We have corrected the sentences that were wrong
-Results section, page 8, lines 200-202: Authors should better explain the following sentence “For TGFβR2 mRNA levels in patients with dilatation, we observed [Control 96.3 (0.05-282.02) vs MFS [4 (0.40-41.3) p=0.05]. But not found differences between MFS and controls in dissection”
We have eliminated unnecessary information and clarified the whole idea in general. Thank you very much for your comments.
Comments on the Quality of English Language
The English language should be revised in the given sentences
We have conducted a thorough English language revision of the entire manuscript to ensure clarity, correctness, and readability.
- Soto